# The Influence of Vitamin D Status on Cognitive Ability in Patients with Bipolar Disorder and Healthy Controls

**DOI:** 10.3390/nu15194111

**Published:** 2023-09-22

**Authors:** Bernadette Leser, Nina Dalkner, Adelina Tmava-Berisha, Frederike T. Fellendorf, Human-Friedrich Unterrainer, Tatjana Stross, Alexander Maget, Martina Platzer, Susanne A. Bengesser, Alfred Häussl, Ina Zwigl, Armin Birner, Robert Queissner, Katharina Stix, Linda Wels, Elena M. D. Schönthaler, Melanie Lenger, Andreas R. Schwerdtfeger, Sieglinde Zelzer, Markus Herrmann, Eva Z. Reininghaus

**Affiliations:** 1Department of Psychology, University of Graz, 8010 Graz, Austria; bernadette.leser@gmail.com (B.L.); andreas.schwerdtfeger@uni-graz.at (A.R.S.); 2Department of Psychiatry and Psychotherapeutic Medicine, Medical University Graz, 8036 Graz, Austria; adelina.tmava@medunigraz.at (A.T.-B.); frederike.fellendorf@medunigraz.at (F.T.F.); tatjana.stross@medunigraz.at (T.S.); alexander.maget@medunigraz.at (A.M.); martina.platzer@medunigraz.at (M.P.); susanne.bengesser@medunigraz.at (S.A.B.); alfred.haeussl@medunigraz.at (A.H.); ina.zwigl@medunigraz.at (I.Z.); armin.birner@kabeg.at (A.B.); robert.queissner@medunigraz.at (R.Q.); katharina.stix@medunigraz.at (K.S.); elena.schoenthaler@medunigraz.at (E.M.D.S.); melanie.lenger@medunigraz.at (M.L.); eva.reininghaus@medunigraz.at (E.Z.R.); 3Faculty of Psychotherapy Science, Sigmund Freud University, 1020 Vienna, Austria; 4Clinical Institute of Medical and Chemical Laboratory Diagnostics, Medical University of Graz, 8036 Graz, Austria; sieglinde.zelzer@medunigraz.at (S.Z.); markus.herrmann@medunigraz.at (M.H.)

**Keywords:** bipolar disorder, vitamin D, functional vitamin D deficiency, 25(OH)D, 24,25(OH)2D, VMR

## Abstract

Recent evidence on the association between vitamin D and cognition in mentally healthy individuals is inconsistent. Furthermore, the link between vitamin D and cognitive ability in individuals with bipolar disorder has not been studied yet. Thus, we aimed to investigate the association between 25-hydroxyvitamin D (25(OH)D), 24,25 dihydroxyvitamin D (24,25(OH)2D, the vitamin D metabolite ratio (VMR) and cognition in a cohort of euthymic patients with bipolar disorder. Vitamin D metabolites were measured simultaneously by liquid-chromatography tandem mass-spectrometry in serum samples from 86 outpatients with bipolar disorder and 93 healthy controls. Neither the inactive precursor 25(OH)D, nor the primary vitamin D catabolite 24,25(OH)2D, or the vitamin D metabolite ratio were significantly associated with the domains “attention”, “memory”, or “executive function” in individuals with bipolar disorder and healthy controls. Further, no vitamin D deficiency effect or interaction group × vitamin D deficiency was found in the cognitive domain scores. In summary, the present study does not support vitamin D metabolism as a modulating factor of cognitive function in euthymic BD patients. Considering the current study’s cross-sectional design, future research should expand these results in a longitudinal setting and include additional aspects of mental health, such as manic or depressive symptoms, long-term illness course and psychopharmacological treatment.

## 1. Introduction

Bipolar disorder (BD) is a serious mental illness marked by pathological mood changes. Depressive and (hypo)manic, as well as euthymic phases occur at an unpredictable sequence. Lifetime prevalence ranges from 0.3–1.5% for bipolar type I disorder and up to 5.5% for bipolar type II disorder with a high heritability [1]. Individuals who suffer from BD are often affected by impaired psychosocial functioning, cardiovascular and metabolic comorbidities, and earlier mortality. Importantly, many patients also exhibit cognitive deficits, which are associated with low everyday functioning [1].

In euthymic patients with BD, the largest effects on cognitive impairment are found in the domains of “memory”, “attention”, and “executive function” [2]. Different sociodemographic, genetic, and clinical factors including psychotic symptoms [3], episode frequency [4,5], somatic conditions such as obesity [6] and metabolic syndrome [7], are assumed to contribute to cognitive deficits. In addition, lifestyle and nutrition can influence the illness course [8]. Therefore, the potential role of vitamin D on cognition in mental disorders has gained significant interest in the recent past [9,10].

Vitamin D, which exists as skin-derived vitamin D3 (cholecalciferol) and food-derived vitamin D2 (ergocalciferol), is converted in the liver to 25-hydroxycholecalciferol (25(OH)D3) and 25-hydroxyergocalciferol (25(OH)D2), respectively. If not stated otherwise, laboratory tests for 25(OH)D capture both forms and thus report total 25(OH)D. Despite being the most abundant vitamin D metabolite in blood, 25(OH)D is still inactive and represents the body’s vitamin D reservoir. Biologically active 1,25-dihydroxycholecalciferol (1,25(OH)2D) is produced on demand via a second hydroxylation in position 1 that is catalyzed by 1α-hydroxylase (CYP27B1). This primarily takes place within the kidneys, with fewer occurrences observed in various other tissues, such as the gut, breast, skeletal muscle, immune cells, and brain. Similar to the expression of 1α-hydroxylase, the vitamin D receptor (VDR) is expressed in a broad range of tissues alluding to pleiotropic effects of vitamin D. In the brain, VDR has been detected in regions that are involved in cognitive processes [10]. Excess amounts of 25(OH)D and 1,25(OH)2D are eliminated by an additional hydroxylation in position 24. The resulting 24-hydroxylated metabolites are further processed and ultimately eliminated from the body. Overall, 24,25(OH)2D is the most abundant vitamin D catabolite that can be measured in blood [11]. 

Individuals with mental illnesses show a significant incidence of vitamin D deficiency [12,13]. For example, Rihal et al. [14] illuminate the associations between vitamin D and schizophrenia, autism, depression, and ADHD. As a result, a deficiency in vitamin D could potentially emerge as a contributing factor to the onset of these neuropsychiatric disorders. Today, it is well accepted that the metabolic actions of vitamin D go far beyond calcium-phosphate homeostasis and involve the brain as well as many other tissues [15]. In their extensive review, Cui et al. [16] provide a description of vitamin D in the central nervous system: Vitamin D supports myelination as well as functional recovery, and alterations in vitamin D levels within brain cells have been linked to a decrease in the density of nerve fibers, delayed and diminished development of dopamine neurons, and impaired release of gamma-aminobutyric acid and glutamate. Additionally, vitamin D is tied to the nerve growth factor, the restoration of dopamine levels, and stress responsiveness. Consequently, vitamin D can be regarded as a significant steroid in the physiology of brain neurons. Therefore, vitamin D contributes to mental health disorders and cognitive functions.

However, existing studies, which investigated potential associations between 25(OH)D and cognitive performance, show inconsistent results. Some studies indicate reduced cognitive performance in individuals with serum 25(OH)D concentrations below the recommended range [17,18], while others did not show any associations between those variables [19,20]. One possible explanation for the inconsistent results is the variable analytical performance of widely used immunoassays for 25(OH)D, which lack sensitivity and specificity [21]. Isotope dilution liquid chromatography tandem mass spectrometry (LC-MS/MS) is the preferred technique for the measurement of 25(OH)D and other related metabolites, which offers high sensitivity and specificity. Furthermore, LC-MS/MS enables the concurrent measurement of various vitamin D metabolites, offering dynamic insights into vitamin D metabolism [22]. A recent study has shown that the parallel measurement of 25(OH)D and 24,25(OH)2D, and calculation of the vitamin D metabolite ratio (VMR), is more specific to establish functional vitamin D deficiency than the isolated measurement of the inactive prohormone 25(OH)D [23]. Mayne et al. [9] provide an overview of the important roles that vitamin D plays in the body and brain, which are also related to cognitive functions. Among others, vitamin D is associated with a mechanism of synaptic plasticity [24]. Synaptic plasticity describes the ability to form new synapses, thus representing a fundamental aspect in learning and memory [25].

### Aim of the Study

The primary objective of this study was to explore the relationship between 25(OH)D, 24,25(OH)2D and VMR with cognitive function in patients with BD and a control group of healthy controls. It was hypothesized that patients with BD and healthy controls with a lower vitamin D status show lower performance in “attention”, “memory”, and “executive function” than those with a higher vitamin D status.

## 2. Materials and Methods

### 2.1. Participants and Study Design

Individuals included in this study were participants of the longitudinal BIPLONG (“bipolar disorder in the long-term”) cohort at the Division of Psychiatry and Psychotherapeutic Medicine of the Medical University of Graz (Austria). BIPLONG aims to investigate the relationship between lifetime psychiatric history and different variables, such as biological parameters, brain function, and lifestyle in the long term. For a detailed description of the study design and other results, we refer to previous publications [7,26].

BD was diagnosed by a psychiatrist or psychologist using the DSM-IV Structured Clinical Interview [27]. BD I and BD II patients were included, but this distinction was not considered in further analyses. Exclusion criteria were neurodegenerative disorders, a premorbid IQ < 70 and the intake of vitamin D supplements. The present study only included data from euthymic patients with BD and mentally healthy controls without a history of a psychiatric disorder life-time and among first-degree relatives. Euthymia was verified by a Hamilton Depression Scale (HAMD; [28]) score of ≤9 and Young Mania Rating Scale (YMRS; [29]) score of ≤8, which were rated by a psychiatrist or psychologist. Only euthymic patients were included to exclude possible effects caused by a manic or depressive episode, since Vrabie et al. [5] showed that patients in a manic episode have higher deficits in certain cognitive domains than depressive or euthymic patients. The study received approval from the ethics committee of the Medical University of Graz (EC-number: 25-335 ex 12/13).

From 635 participants of the BIPLONG study, 349 patients were diagnosed with BD. In total, 330 participants (180 patients with BD) were excluded due to missing vitamin D results. Further, information on current mood status was not available for 14 patients with BD, and 50 patients with BD were excluded because they had a YMRS score > 8 and/or a HAMD score > 9 and were therefore not considered euthymic. Cognitive data were missing for 57 participants (15 patients with BD). Four BD patients used vitamin D supplements. One healthy control participant was excluded due to impossible test values. Finally, 86 patients with BD (47 men, 39 women) and 93 healthy controls (29 men, 64 women) were included in the present study.

### 2.2. Determination of Vitamin D Metabolites

Fasting blood samples were collected for vitamin D testing and the measurement of numerous other blood biomarkers that are not subject of this study. Vitamin D status was assessed by measuring 25(OH)D and 24,25(OH)2D3 in serum. The results were used to calculate the VMR. Both vitamin D metabolites were measured simultaneously with a fully validated LC-MS/MS method that has performed satisfactorily for several years in the Vitamin D External Quality Assurance Scheme (DEQAS) and is regularly controlled by stringent internal quality control procedures. With the method used here, serum 25(OH)D represents the sum of 25(OH)D3 and 25(OH)D2, which are captured separately. Although both forms of 25(OH)D are catabolized by 24-hydroxylation, 25(OH)D2 normally accounts for 1–3% of the total 25(OH)D resulting in 24,25(OH)D2 concentrations below the limit of quantitation. Therefore, vitamin D catabolism is assessed by 24,25(OH)2D3 as the only catabolite that is measurable with the method used here.

According to current recommendations from the Institute of Medicine (IOM), a 25(OH)D concentration < 50 nmol/L is considered deficient [30]. Furthermore, functional vitamin D deficiency was diagnosed using a recently published approach from Herrmann [31]. This approach requires a 24,25(OH)2D3 concentration < 3 nmol/L in combination with a VMR < 4%. Individuals that fulfilled only one of these criteria were classified as having a non-optimal vitamin D metabolism. For more details, see Zelzer et al. [22].

### 2.3. Assessment Scales

Cognitive performance of the participants was assessed using an extensive cognitive test battery that included the Trail Making Test Part A/B [32], the Color-Word Interference Test by J. R. Stroop [33], and the California Verbal Learning Test (CVLT; [34]). All tests were provided in the German language. The raw scores were summed up to three cognitive domain scores: “attention”, “memory”, and “executive function”. For this purpose, the TMT-A [32] and the Color and Word part of the Stroop test [33] were combined into the domain score “attention”. The subscales of the CVLT [34] provided information about the domain score “memory”. The interference part of the Stroop test [33] as well as part B of the TMT [32] were used to measure “executive function”. Additionally, the premorbid IQ was measured with the Multiple-Choice Vocabulary Intelligence Test (MWT-B; [35]).

### 2.4. Statistics

*Z*-transformations were performed in order to sum up the cognitive raw scores to domain scores. Reaction times were inverted so that higher scores reflect higher performance in all domains. Since normal distribution was not given for the domain scores, a Rankit-transformation of these scores was performed.

Multiple hierarchical regression analyses were calculated to investigate whether 25(OH)D, 24,25(OH)2D3, or VMR were associated with the cognition domain scores in BD patients and healthy controls. The independent variables were 25(OH)D, 24,25(OH)2D3, and VMR. Premorbid IQ and age were included as control variables. The following calculations were made to verify the prerequisites for conducting a multiple hierarchical regression analysis: Correlations and chi-square tests for linearity between independent and dependent variables, Durbin–Watson coefficients for autocorrelations, variance of inflation factors (VIF), tolerance values for multicollinearity, scatterplots for homoscedasticity of residuals, and the Shapiro–Wilk test for testing the normal distribution of the residuals. These prerequisites were met in all regression analyses.

To investigate if patients with BD versus healthy controls with or without functional vitamin D deficiency show a difference in the cognitive domain scores, a two-way multivariate analysis of covariance (MANCOVA) was calculated. For this purpose, the vitamin D catabolite 24,25(OH)2D3 and the VMR were considered together to classify the functional vitamin D deficiency. Premorbid IQ and age were included as covariables in the model. Skewness and kurtosis for normal distribution, Levene’s test for homogeneity, and graphs for linearity were calculated to verify the prerequisites of the MANCOVA. All prerequisites were met.

## 3. Results

### 3.1. Demographics and Sample Characteristics

The mean age of individuals with BD was 45.17 ± 13.15 years (min = 18 years, max = 72 years). The mean age of the healthy controls was 37.75 ± 15.32 years (min = 19 years, max = 76 years). Age, premorbid IQ, YMRS score, HAMD score, 25(OH)D, 24,25(OH)2D3, VMR, the cognitive domain scores, and the differences (as calculated by Mann–Whitney *U*-tests and chi-square tests) between patients with BD and the healthy controls can be seen in Table 1.

### 3.2. Multiple Hierarchical Regressions

Three multiple hierarchical regression analyses were calculated to investigate whether there is an association between 25(OH)D, 24,25(OH)2D as well as VMR with the three domain scores “attention”, “memory”, and “executive function” in individuals with BD and healthy controls. In the first model, age was included; in the second, age and premorbid IQ; in the third model, age, premorbid IQ, and 25(OH)D; in the fourth model, age, premorbid IQ, 25(OH)D, and 24,25(OH)2D3; and in the fifth model, all variables (age, premorbid IQ, 25(OH)D, 24,25(OH)2D3, and VMR) were included. The order of entering the predictors was based on considerations of causal priority and served the purpose of controlling for potential effects.

### 3.3. Patients with Bipolar Disorder

#### 3.3.1. Attention

Multiple hierarchical regression analysis revealed no significant association between the individual level of 25(OH)D 24,25(OH)2D3 or VMR and attention (Model 1: F(1, 84) = 7.31, *p* = 0.008; Model 2: F(2, 83) = 4.82, *p* = 0.010; Model 3: F(3, 82) = 3.73, *p* = 0.014; Model 4: F(4, 81) = 3.28, *p* = 0.015; Model 5: F(5, 80) = 2.68, *p* = 0.027). In all steps, only age showed a significant effect on attention. Table 2 shows the regression coefficients.

#### 3.3.2. Memory

The results of the multiple hierarchical regression analysis indicated no significant association between 25(OH)D, 24,25(OH)2D3 or VMR and “memory”. In all steps (Model 1: F(1, 84) = 8.03, *p* = 0.006; Model 2: F(2, 83) = 9.83, *p* < 0.001; Model 3: F(3, 82) = 6.82, *p* < 0.001; Model 4: F (4, 81) = 5.06, *p* = 0.001, Model 5: F(5, 80) = 4.20, *p* = 0.002), age showed a significant effect on “memory”, as well as premorbid IQ once it was taken into the model (see Table 2).

#### 3.3.3. Executive Function

In all steps of the multiple hierarchical regression analysis, age showed a significant effect on “executive function” (Model 1: F(1, 84) = 11.48, *p* = 0.001). Moreover, premorbid IQ showed a significant effect on “executive function” once it was taken into the model (Model 2: F(2, 83) = 10.53, *p* < 0.001; Model 3: F (3, 82) = 7.01, *p* < 0.001; Model 4: F(4, 81) = 5.78, *p* < 0.001; Model 5: F(5, 80) = 4.83, *p* < 0.001). No significant associations between 25(OH)D, 24,25(OH)2D3, or VMR and “executive function” were found (see Table 2).

#### 3.3.4. Healthy Controls

The results from hierarchical regression analyses in the healthy controls did not differ from patients in BD and results can be seen in the Appendix A.

#### 3.3.5. Multivariate Analysis of Covariance

A MANCOVA was calculated to test for differences between patients with BD and the healthy controls (factor group) with functional vitamin D deficiency versus those without (factor vitamin D deficiency) in the cognitive domain scores “attention”, “memory”, and “executive function”. The covariates were the participants’ age and their premorbid IQ, which was calculated with the MWT-B [35]. The descriptive results for vitamin D deficiency in both groups are listed in Table 3, showing that 43% of patients and 33.3% of controls were in the non-optimal group. MANCOVA showed a significant effect of group (F(3171) = 2.67, *p* = 0.049, η_p_^2^ = 0.05). The main effect for vitamin D deficiency (F(3171) = 0.47, *p* = 0.703, η_p_^2^= 0.01) and the interaction effect for group × vitamin D deficiency (F(3171) = 0.99, *p* = 0.542, η_p_^2^ = 0.01) were not significant. Age (F(3171) = 24.23, *p* < 0.001, η_p_^2^= 0.30) and premorbid IQ (F(3171) = 10.71, *p* < 0.001, η_p_^2^ = 0.16) were significant covariates. Univariate post hoc analyses of variance (ANOVA) showed that BD patients exhibited lower performance in “memory” (F(1173) = 4.63, *p* = 0.033, η_p_^2^ = 0.03) and “executive function” (F(1173) = 5.53, *p* = 0.020, η_p_^2^ = 0.03) than the healthy controls. No difference between the two groups was found for the domain score attention (F(1173) = 3.67, *p* = 0.057, η_p_^2^ = 0.21).

## 4. Discussion

The present study investigated whether the vitamin D status (25(OH)D), the main vitamin D catabolite (24,25(OH)2D3), or the vitamin D metabolite ratio (VMR) have an effect on cognition (divided into “attention”, “memory”, and “executive function”) in patients with BD. The findings do not substantiate a significant relationship between vitamin D metabolism and cognitive function. The different indices of vitamin D metabolism were not associated with “attention”, “memory”, or “executive function”, neither in BD patients nor in healthy controls. Our results are contrary to previous data, which showed reduced memory function in older individuals with low 24,25(OH)2D3 concentrations [36]. Besides vitamin D, other factors that influence cognition may also be of interest. For example, associations between cognitive deficits and elevated triacylglycerol and glucose [37] or higher cytokine IL6 [38] have already been found. Other factors, such as the metabolic syndrome [7] anthropometric parameters also showed an association with cognition in patients with BD. Recent research emphasizes the importance of utilizing vitamin D metabolites to determine the vitamin D status, especially to detect deficiency [11,22].

In addition, the occurrence of functional vitamin D deficiency was comparable in BD patients and healthy controls. Therefore, 24,25(OH)2D3 and VMR were used to determine functional vitamin D deficiency. However, no functional vitamin D deficiency effect could be found in our analyses, which tested differences in cognitive domain scores between patients with BD and the healthy controls. In concordance with the current literature, we found that individuals with BD showed poorer memory and executive function performance than the healthy controls. Our findings align with prior research, in which patients with BD also showed cognitive deficits in memory [2], executive function and processing speed [3].

The inconsistent results could be, at least partially, explained by the substantially different age of the participants. In BIPLONG, the participants were rather young, which limits the chance for significant effects induced by vitamin D deficiency. The absence of an association between vitamin D and the domain scores “attention”, “memory”, and “executive function” argues against vitamin D deficiency as a promotor of cognitive dysfunction in BD. This is an important finding since BD patients are often severely impaired by cognitive and somatic deficits. Further possible reasons for inconsistent results regarding an association between BD and poorer cognitive performance could be the use of different cognitive tests or different assignments of these tests to the cognitive domain scores.

A particular strength of our study is the measurement of 25(OH), 24,25(OH)2D3 and VMR, which provides a more comprehensive insight into vitamin D metabolism [11,22]. Determining 24,25(OH)2D3 and VMR offers important metabolic information beyond the sole measurement of 25(OH)D. Functional vitamin D deficiency was equally prevalent in BD patients and in the healthy controls, which argues against a significant role of vitamin D for cognition. However, mean 25(OH)D concentrations of 56.37 nmol/L and 57.29 nmol/L were found in BD patients and the controls, respectively, suggesting a relatively adequate vitamin D supply, which limits the margin for significant effects. In contrast to the present study, previous investigations mainly included hospitalized patients with severe symptoms, in whom 25(OH)D concentrations are typically lower [4]. Existing evidence [4] suggests that BD patients with more frequent manic episodes perform worse on the CVLT than those with less frequent manic episodes. This is an important aspect as manic patients are more often affected by worse memory and executive function compared to depressed and euthymic patients. Therefore, it remains plausible that there is a connection between vitamin D and cognition in symptomatic BD patients. Furthermore, it could be theorized that vitamin D supplementation might have positive effects on mood in BD patients that are mediated via the serotonergic pathway [39]. This hypothesis finds support in data from healthy individuals, which demonstrate a beneficial impact of vitamin D supplementation on mood [39]. However, in a recent literature review, Huiberts et al. [40] concluded that the existing data are inconsistent.

## 5. Limitations of the Study

Our study possesses various strengths and limitations that warrant attention when interpreting the findings. The simultaneous determination of 25(OH)D, 24,25(OH)2D3 and VMR provides important metabolic information that extends beyond the evaluation of vitamin D stores by only measuring 25(OH)D. These parameters offered the possibility to establish functional vitamin D deficiency, which has been linked to adverse clinical outcomes [31]. Furthermore, the use of a validated state of the art LC-MS/MS method ensured accurate results that are not affected by the analytical limitations of commonly used 25(OH)D immunoassays [21]. Another advantage of this study is a rather large cohort of euthymic BD patients in whom the diagnosis of BD was thoroughly established and cognition was assessed by a well-elaborated cognitive test battery. However, when considering the potential impacts of vitamin D on sleep and mood, significant associations between vitamin D and cognition in manic or depressive BD patients cannot be excluded. Also, many participants were treated with psychopharmacological medication so that potential drug effects on vitamin D metabolism and cognitive function cannot be ruled out. Notably, patients with BD were taking different medications, making it difficult to identify the different effects of each medication. Previous studies reported heterogeneous cognitive effects of commonly used medications in BD patients [41,42]. For example, Paterson et al. [41] found no effect of lithium on attention, but a positive effect on psychomotor speed in patients with BD, whereas Holmes et al. [42] reported a negative effect on attention and affective processing. Factors such as medications, organic diseases, diet, and sun exposure are potential confounders of vitamin D metabolite concentrations, and the absence of this information weakens the statistical power of the present study. However, to reduce confounding factors, we excluded subjects with vitamin D supplementation. In addition, the known seasonal variations of 25(OH)D were accounted for by distributing the study dates relatively evenly throughout the year. Furthermore, the sex-specific differences in the concentration of vitamin D in patients with severe mental illness could be relevant to explore the potentially sex-specific involvement of vitamin D in the pathogenesis of mental health disorders. Previous research suggests a female-specific involvement of vitamin D in the pathogenesis of depression, relating vitamin D to the production/release of gonadal hormones [43,44]. This view has been supported by previous research providing evidence that the gender difference in the prevalence of depression is less evident during menopause, when gonadial hormonal flux stabilizes [45].

Another aspect to consider is the unknown duration of vitamin D deficiency that may vary between the study participants. When taking into account that vitamin D deficiency has potential adverse effects on the signaling pathways in the brain, a longer duration of vitamin D deficiency could potentially be more relevant for cognition than short-lasting deficiencies. As vitamin D3 production is also subject to pronounced seasonal variation, the time of blood collection may also be a factor [46]. Finally, this study included only euthymic patients so that potential effects of manic or depressive symptoms could not be assessed. Ultimately, this study is a cross-sectional study, which means that causal relationships cannot be assessed.

Further studies, ideally of a prospective nature, are needed to consolidate the present results. Such studies should include symptomatic and asymptomatic patients to reveal possible effects released by a manic or depressive episode. In addition, it cannot be excluded that vitamin D influences cognition indirectly through the modulatory effects of other metabolic pathways and inflammatory processes.

## 6. Conclusions

In summary, the present study does not support vitamin D metabolism as a modulating factor of cognitive function in euthymic BD patients. Moreover, the incidence of low vitamin D stores and functional vitamin D deficiency did not significantly differ between BD patients and healthy controls. As this is the first study that addressed the impact of vitamin D on cognitive function in BD patients, more research is needed to expand the present findings. Specifically, longitudinal studies that include euthymic and symptomatic BD patients are of particular interest.

## Figures and Tables

**Table 1 nutrients-15-04111-t001:** Sample characteristics and differences between patients with BD and healthy controls.

	Patients with BD (*n =* 86)	Healthy Controls (*n* = 93)	Differences
Age (Years), M (±SD)	45.17 (13.15)	37.75 (15.33)	U = 2732.00 **, Z = −3.66
Sex (*n*, %)			χ^2^ = 10.07 *, φ = 0.24
Male Female	47 (54.7%) 39 (45.3%)	29 (31.2%) 64 (68.8%)	
Premorbid IQ, M (±SD)	109.51 (15.23)	112.18 (14.71)	U = 3579.50, Z = −1.22, *p* = 0.224
YMRS, M (±SD)	0.93 (1.88)	0.12 (0.55)	U = 3082.50 **, Z = −4.13
HAMD, M (±SD)	4.63 (3.45)	0.27 (0.95)	U = 902.50 **, Z = −9.78
Vitamin D variables, (*n*, %)			
25(OH)D, M (±SD)	56.37 (23.64) nmol/L	57.29 (23.95) nmol/L	U = 3813.50, Z = −0.54
24,25(OH)_2_D_3_, M (±SD)	3.59 (2.11) nmol/L	4.02 (2.51) nmol/L	U = 3608.50, Z = −1.13
VMR, M (±SD)	6.31 (2.08) %	6.91 (2.51) %	U = 3465.00, Z = −1.54
25(OH)D			χ^2^ = 0.32, φ = 0.04, *p* = 0.570
<50 nmol/L >50 nmol/L	35 (40.7%) 51 (59.3%)	34 (36.6%) 59 (63.4%)	
24,25(OH)_2_D_3_			χ^2^ = 1.39, φ = 0.09, *p* = 0.239
<3 nmol/L >3 nmol/L	36 (41.9%) 50 (58.1%)	31 (33.3%) 62 (66.7%)	
VMR (*n*, %)			χ^2^ = 0.05, φ = 0.02, *p* = 0.829
<4% >4%	13 (15.1%) 73 (84.9%)	13 (14.0%) 80 (86.0%)	
Cognitive variables, M (±SD)			
TMT-A (s)	35.21 (11.98)	26.04 (9.74)	U = 2082.50 **, Z = −5.53
TMT-B (s)	90.99 (51.14)	57.12 (20.96)	U = 2036.00 **, Z = −5.67
Stroop color word reading (s)	33.15 (7.06)	29.06 (4.63)	U = 2465.00 **, Z = −4.43
Stroop color naming (s)	51.70 (9.57)	44.11 (6.92)	U = 2051.00 **, Z = −5.62
Stroop interference (s)	85.16 (24.33)	67.67 (14.73)	U = 2054.00 **, Z = −5.62
CVLT trial 1–5	49.74 (13.26)	60.48 (11.80)	U = 2141.00 **, Z = −5.37
CVLT short delay free recall	10.02 (3.51)	12.82 (2.96)	U = 2090.00 **, Z = −5.54
CVLT short delay cued recall	11.03 (3.31)	13.27 (2.66)	U = 2363.00 **, Z = −4.76
CVLT long delay free recall	10.58 (3.57)	13.24 (3.09)	U = 2182.50 **, Z = −5.28
CVLT long delay cued recall	11.34 (3.29)	13.46 (2.91)	U = 2390.00 **, Z = −4.68

Note: Results from Mann–Whitney U-tests or chi-square tests. IQ = Intelligence quotient, YMRS = Young Mania Rating Scale, HAMD = Hamilton Depression Scale, CVLT = California Verbal Learning Test, TMT = Trail Making Test; ** *p* < 0.01; * *p* < 0.05.

**Table 2 nutrients-15-04111-t002:** Association of age, premorbid IQ, 25(OH)D, 24,25(OH)_2_D_3_, VMR with “attention”, “memory”, and “executive function” in patients with BD.

		Attention					Memory					Executive Function				
		B	95%CI	β	*t*	*p*	B	95%CI	β	*t*	*p*	B	95%CI	β	*t*	*p*
Model 1	Age	−0.02	[−0.04, −0.01]	−0.28	−2.70	**0.008**	−0.02	[−0.04, −0.01]	−0.30	−2.83	**0.006**	−0.03	[−0.04, −0.01]	−0.35	−3.39	**0.001**
Model 2	Age	−0.02	[−0.04, −0.01]	−0.32	−3.01	**0.003**	−0.03	[−0.04, −0.01]	−0.38	−3.71	**<0.001**	−0.03	[−0.05, −0.02]	−0.42	−4.15	**<0.001**
	Premorbid IQ	0.01	[−0.00, 0.02]	0.16	1.49	0.140	0.02	[0.01, 0.04]	0.33	3.27	**0.002**	0.02	[0.01, 0.03]	0.30	2.93	**0.004**
Model 3	Age	−0.2	[−0.04, −0.01]	−0.31	−2.86	**0.005**	−0.03	[−0.04, −0.01]	−0.37	−3.58	**<0.001**	−0.03	[−0.05, −0.02]	−0.42	−4.06	**<0.001**
	Premorbid IQ	0.01	[−0.01, 0.02]	0.14	1.34	0.185	0.02	[0.01, 0.03]	0.32	3.13	0.002	0.02	[0.01, 0.03]	0.29	2.84	0.006
	25(OH)D	−0.26	[−0.68, 0.16]	−0.13	−1.22	0.226	−0.19	[−0.59, 0.22]	−0.09	−0.92	0.359	−0.09	[−0.49, 0.32]	−0.04	−0.42	0.674
Model 4	Age	−0.02	[−0.04, −0.01]	−0.31	−2.91	**0.005**	−0.03	[−0.04, −0.01]	−0.37	−3.56	**<0.001**	−0.03	[−0.05, −0.02]	−0.42	−4.11	**<0.001**
	Premorbid IQ	0.01	[−0.01, 0.02]	0.13	1.15	0.252	0.02	[0.01, 0.04]	0.32	3.09	**0.003**	0.02	[0.00, 0.03]	0.27	2.65	**0.010**
	25(OH)D	−0.05	[−0.57, 0.47]	−0.03	−0.20	0.841	−0.19	[−0.69, 0.31]	−0.09	−0.75	0.453	0.11	[−0.38, 0.61]	0.06	0.46	0.649
	24,25(OH)_2_D_3_	−0.35	[−0.87, 0.17]	−0.17	−1.35	0.181	0.01	[−0.50, 0.50]	0.01	0.020	0.985	−0.34	[−0.83, 0.15]	−0.17	−1.37	0.176
Model 5	Age	−0.02	[−0.04, −0.01]	−0.31	−2.89	**0.005**	−0.03	[−0.04, −0.01]	−0.37	−3.54	**<0.001**	−0.03	[−0.05, −0.02]	−0.42	−4.10	**<0.001**
	Premorbid IQ	0.01	[−0.01, 0.02]	0.13	1.18	0.240	0.02	[0.01, 0.04]	0.33	3.13	0.002	**0.02**	[0.01, 0.03]	0.28	2.70	**0.008**
	25(OH)D	−0.03	[−0.56, 0.49]	−0.02	−0.13	0.899	−0.16	[−0.67, 0.34]	−0.08	−0.64	0.523	0.14	[−0.35, 0.64]	0.07	0.57	0.569
	24,25(OH)_2_D_3_	−0.29	[−0.85, 0.27]	−0.14	−1.02	0.313	0.10	[−0.44, 0.64]	0.05	0.36	0.722	−0.24	[−0.77, 0.30]	−0.12	−0.89	0.375
	VMR	−0.21	[−0.87, 0.46]	−0.08	−0.63	0.534	−0.29	[−0.93, 0.35]	−0.10	−0.91	0.364	−0.32	[−0.95, 0.31]	−0.12	−1.01	0.315

Note: Attention: Model 1: R^2^ = 0.08, R^2^corr = 0.07, Model 2: R^2^ = 0.10, R^2^corr = 0.08, Model 3: R^2^ = 0.12, R^2^corr = 0.09, Model 4: R^2^ = 0.14, R^2^corr = 0.10, Model 5: R^2^ = 0.14, R^2^corr = 0.09; Memory: Model 1: R^2^ = 0.09, R^2^corr = 0.08, Model 2: R^2^ = 0.19, R^2^corr = 0.17, Model 3: R^2^ = 0.20, R^2^corr = 0.17, Model 4: R^2^ = 0.20, R^2^corr = 0.16, Model 5: R^2^ = 0.21, R^2^corr = 0.16; Executive Function: Model 1: R^2^ = 0.12, R^2^corr = 0.11, Model 2: R^2^ = 0.20, R^2^corr = 0.18, Model 3: R^2^ = 0.20, R^2^corr = 0.18, Model 4: R^2^ = 0.22, R^2^corr = 0.18, Model 5: R^2^ = 0.23, R^2^corr = 0.18. Bold printed *p*-values are significant.

**Table 3 nutrients-15-04111-t003:** Percentage of optimal and non-optimal functional vitamin D status in individuals with BD and healthy controls.

	Patients with BD (*n =* 86)	Healthy Controls (*n* = 93)	
Optimal	49 (57.0%)	62 (66.7%)	χ^2^ = 1.78, φ = 0.10, *p* = 0.182
Non-optimal	37 (43.0%)	31 (33.3%)	

Note: BD = Bipolar disorder; Optimal = 24,25(OH)2D3 > 3 and VMR > 4; Non-optimal = 24,25(OH)2D3 < 3 and/or VMR < 4. Results from chi-square tests.

## Data Availability

Not applicable.

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
