# Peer review of "The Influence of Vitamin D Status on Cognitive Ability in Patients with Bipolar Disorder and Healthy Controls"

_nutrients, 2023, doi:10.3390/nu15194111_

Round 1

Reviewer 1 Report

In the present study, the authors investigated the relationship between 25(OH)D, 24,25(OH)2D and the vitamin D metabolite ratio (VMR) with cognitive function in patients with BD and healthy controls. The present results have demonstrated that neither the inactive precursor 25(OH)D, nor the primary vitamin D catabolite 24,25(OH)2D, or the vitamin D metabolite ratio were significantly associated with attention, memory, or executive function in individuals with bipolar disorder and healthy controls and that no vitamin D deficiency effect or interaction group × vitamin D deficiency was found in the cognitive domain scores.

I think the manuscript includes new and intriguing findings, however the authors should revise it according to the following concerns;

The authors should discuss more in detail on the role of vitamin D on the pathology of neuropsychiatric disorders, citing relevant recent literatures such as Cui X et al., Nutrients 2022 PMID: 36297037 and Rihal V et al., Psychiatric Research 2022 PMID: 36049434. Thereafter, the authors should discuss more in detail on the putative mechanism how vitamin D has an effect on cognitive function in patients with BD, citing relevant literatures.

fair

Author Response

We totally agree. We revised accordingly to your suggestions, please see page 2 and 3. And, please see the point-to-point response attached. 

Reviewer 2 Report

This study explores the influence of vitamin D status on cognitive ability in patients with bipolar disorder vs. healthy controls. Taking into account the limitations acknowledged by the Authors, the methodology of the study is appropriate to the objectives and the investigational process well supports the results. Please refer to the following observations:

Inclusion criteria: Were diagnoses of both BD I and BD II allowed? If this distinction was made during the screening phase, was any analysis conducted in patients with type I vs. type II BD?

Exclusion criteria are insufficient because there are also medications or organic diseases that could reduce the level of vitamin D. At least, this aspect should be mentioned in the “Discussion” section.

Lines 152-155, 196, - please consider writing attention, executive dysfunction, and memory between quotation marks, since they are specific domains constructed for this study.

Author Response

We totally agree and have revised everything accordingly to your suggestions. please see the point-to-point response attached. 

Reviewer 3 Report

This is a very interesting paper investigating the influence of Vitamin D on cognitive ability in patients suffering from bipolar disorder compared with healthy controls. The paper is well-written, and of interest for the journal; however, several minor changes are recommended before considering its publication.

ABSTRACT.

1-I recommend to summarize the first part of the abstract. The main aims should be also be reported briefly, and the methods are recommended to be expanded. The authors should describe the study design and how participants were recruited (outpatients vs inpatients??).

2- I do not recommend to explain limitations and strenghts of the study in the abstract section. I recommend to omit lines 32 to 35, and draw conclusions derived from the results.

INTRODUCTION

1- Vitamin D can be reduced in several mental illnesses, other than bipolar disorder. I recommend to build a brief paragraph introducing the topic of vitamins in mental disorders, and particularly, vitamin D in SMI.

2- The main aims can be described in a separate subsection (1.1).

METHODS

1. I recommend to number the subsections of the Methods section.

2. The first subsection should be renamed as Participants and Study design.

3. Vitamin D metabolites is refering to laboratory tests. Please, rename this section.

4. Psychometric instruments used to evaluate cognition should be explained under a section of "assessment scales".

RESULTS

1-Is there any difference between men and women in vitamin D levels? Is there any interaction between low and normal levels, sex and diagnosis group (patients vs. healthy controls?).

2- Based on "sex " variables, both groups are unbalanced (bipolar disorder and healthy controls). Could it be a limitation of the study?

DISCUSSION

1- I recommend to discuss gender differences in vitamin levels and medical comorbidity in patients with severe mental illness, particularly in vitamine D levels.

2- Is there any potential difference between premenopausal and postmenopausal women? If this can not be analysed, I recommend to include it in the discussion section.

Author Response

Thank you very much for your comments. We have revised the paper according your suggestions which very much helped us to improve the paper´s quality. Please see attached the point-to-point response, all changes are marked yellow. 

Round 2

Reviewer 1 Report

The manuscript has properly been revised according to the reviewers' comments.

none